# Update of PSMA Theranostics in Prostate Cancer: Current Applications and Future Trends

**DOI:** 10.3390/jcm11102738

**Published:** 2022-05-12

**Authors:** Chalermrat Kaewput, Sobhan Vinjamuri

**Affiliations:** 1Department of Radiology, Division of Nuclear Medicine, Faculty of Medicine, Siriraj Hospital, Mahidol University, Bangkok 10700, Thailand; 2Department of Nuclear Medicine, Royal Liverpool University Hospital, Liverpool L7 8XP, UK; sobhan.vinjamuri@gmail.com

**Keywords:** PSMA, prostate cancer, theranostics, PET/CT, radionuclide therapy

## Abstract

There is now an increasing trend for targeting cancers to go beyond early diagnosis and actually improve Progression-Free Survival and Overall Survival. Identifying patients who might benefit from a particular targeted treatment is the main focus for Precision Medicine. Radiolabeled ligands can be used as predictive biomarkers which can confirm target expression by cancers using positron emission tomography (PET). The same ligand can subsequently be labeled with a therapeutic radionuclide for targeted radionuclide therapy. This combined approach is termed “Theranostics”. The prostate-specific membrane antigen (PSMA) has emerged as an attractive diagnostic and therapeutic target for small molecule ligands in prostate cancer. It can be labeled with either positron emitters for PET-based imaging or beta and alpha emitters for targeted radionuclide therapy. This review article summarizes the important concepts for Precision Medicine contributing to improved diagnosis and targeted therapy of patients with prostate cancer and we identify some key learning points and areas for further research.

## 1. Introduction

Prostate cancer (PCa) was the second most common cancer and the fifth leading cause of cancer death among men in 2020, with an estimated 1.4 million new cases and 375,000 deaths worldwide, according to data from the GLOBOCAN database [1]. The 5-year survival rate of locally advanced PCa is nearly 100%, however, in the case of patients presenting with metastatic disease at the time of diagnosis, the rate is significantly lower (31%) [2]. Thus, the accurate diagnosis and staging of new patients with prostate cancer are important. The detection of extraprostatic disease is important in deciding on the most appropriate loco-regional or systemic therapeutic options. Therefore, the development of new strategies for earlier diagnosis, accurate staging, and treatment of metastatic PCa is essential.

According to current guidelines, multiparametric magnetic resonance imaging (mpMRI) is recommended before prostate biopsy [3] and is a major tool for the optimization of prostate biopsies. The use of MRI before biopsy and MRI-targeted biopsy is better than standard transrectal ultrasonography-guided biopsy in men at clinical risk for PCa who had not previously undergone biopsy [4]. The previous study of Eklund M. et al. compared between MRI targeted biopsy and standard biopsy in men with MRI suggested PCa. Their study revealed that MRI-targeted biopsy was noninferior to standard biopsy for detecting clinically significant PCa but resulted in less detection of clinically insignificant PCa [5]. MRI-targeted biopsy has not yet completely replaced standard biopsy; there is usually a combination of both methods.

The meta-analysis of Zhang et al. revealed the diagnostic accuracy of Prostate Imaging Reporting and Data System version 2 (PI-RADS V2) (PI-RADS V2) for PCa detection with mpMRI with a pooled sensitivity of 85% (95% CI: 0.78–0.91), pooled-specificity of 71% (95% CI: 0.60–0.80), pooled positive likelihood ratio of 2.92 (95% CI: 2.09–4.09), pooled negative likelihood ratio of 0.21 (95% CI: 0.14–0.31), and pooled diagnostic odds ratio of 14.08 (95% CI: 7.93–25.01), respectively [6]. The study of Drost F. et al. also revealed similar results with a pooled sensitivity and specificity of 91% and 37% for the International Society of Urological Pathology (ISUP) grade ≥2, and 95% and 35% for ISUP grade ≥3, respectively [7]. Both studies indicated that PI-RADS V2 appears to have good diagnostic accuracy in patients with PCa lesions with high sensitivity and moderate specificity. However, no recommendation regarding the best threshold can be provided because of heterogeneity.

The meta-analysis of Hovel et al. indicates that both computed tomography (CT) and MRI perform equally poorly in the detection of lymph node (LN) metastases from PCa [8] due to the inadequacy of size thresholds in the detection of disease in normal-sized nodes. Bone scintigraphy (BS) has been the historical mainstay of bone metastasis evaluation and has high diagnostic accuracy, primarily reliant on PSA levels, with low sensitivity for metastatic detection, especially at low PSA levels [9]. However, some PCa lesions may still not be detected using cross-sectional imaging as well as isotope BS. In these patients, additional molecular data obtained by PET imaging with PCa-specific tracers can be helpful. Gallium-68 (^68^Ga)--labeled PSMA PET could be used in conjunction with mpMRI for this application to enable fused image-guided biopsies using MRI for anatomical localization and important details from mpMRI to increase the diagnostic performance [10].

Serum prostate-specific antigen (PSA) is a glycoprotein that is expressed by both normal and abnormal prostate cells. It may be elevated in benign prostatic hypertrophy (BPH), prostatitis, and other non-malignant conditions. It is the most commonly used biomarker for PCa screening. The recent guidelines of the European Association of Urology (EAU), European Association of Nuclear Medicine (EANM), European Society for Radiotherapy and Oncology (ESTRO), European Society of Urogenital Radiology (ESUR), and International Society of Geriatric Oncology (SIOG) on prostate cancer [11] strongly recommend early PSA testing to well-informed men at elevated risk of having PCa, including men from 50 years of age, men from 45 years of age with a family history of PCa, men of African descent from 45 years of age, and men carrying BRCA2 mutations from 40 years of age. PSA is considered a reliable marker of recurrent disease after initial treatment as well as a prognostic marker to determine the extent of involvement elsewhere. A higher baseline serum PSA is related with an increased risk of more advanced disease. This information is incorporated into the discussion about prediction of prognosis. The EAU uses a 3-tiered system for risk stratification of localized PCa as described in Table 1 [11].

“Theranostics” refers to the combination of a predictive biomarker with a therapeutic agent and is now an increasing trend for targeting cancers to go beyond early diagnosis and actually improve Progression-Free Survival (PFS) and Overall Survival (OS). The theranostic concept of PCa based on prostate-specific membrane antigen (PSMA) overexpression, led to the use of PSMA ligands for systemic therapy in patients with metastatic PC. PSMA has emerged as an attractive diagnostics PCa and therapeutic target for small molecule ligands in prostate cancer. It can be labeled with either positron emitters for PET-based imaging or beta and alpha emitters for targeted radionuclide therapy.

In this review, we aim to summarize the important concepts for PSMA theranostics in PCa patients and we identify some key learning points and areas for further research.

## 2. Methods

We utilized PubMed and Google Scholar for published studies and clinicals.gov for interested ongoing and completed clinical trials on PSMA theranostics in patients with PCa as of December 2021. MEDLINE databases (Pubmed and Web of Science) were searched using the following keywords: “Prostate Cancer” AND “PSMA targeted therapy”, “Prostate Cancer” AND “PSMA radioligand therapy”, and “Prostate cancer” AND “PSMA treatment”. Although numerous studies were found, mostly larger and recent systematic review, meta-analysis, and randomized controlled trial studies were selected for this review. Clinical trials (e.g., androgen receptor-axis-targeted therapies (ARAT), DNA damage repair inhibitor (PARP-inhibitors), checkpoint inhibitor immunotherapy, alpha-emitting PSMA-targeted radioligand therapy, and anti-PSMA radioimmunotherapy) were also selected. An overview of the selected studies of PSMA-targeted radioligand therapy (RLT) in PCa for this review is provided in Table 2.

## 3. PSMA-Based Imaging

Prostate-specific membrane antigen (PSMA) is a type II transmembrane protein which has a small intracellular and transmembrane portion and a relatively large extracellular portion consisting of 750 amino acids (19-amino-acid intracellular, 24-amino acid transmembrane and 707- amino-acid extracellular portions). The PSMA gene (known as folate-hydrolase activity1; FOLH1) is situated on the short arm of chromosome-11 in a region that is not normally deleted in PCa [24,25,26]. 

PSMA expression has been consistently demonstrated by immunohistochemistry (IHC) and other techniques in normal, hyperplastic prostate tissues and in invasive carcinomas [27,28,29,30]. Within normal prostate cells, PSMA is usually located within the cytoplasm and at the apical aspect of the epithelium surrounding the ducts and, therefore, it is unavailable for binding. Cytoplasmic PSMA is truncated at the N-terminus and is called PSM, it has no FOLH or capacity to hydrolyze N-acetylaspartylglutamic acid [31,32]. Dysplastic and/or neoplastic transformation of the prostatic cell leads to the transfer of PSMA from the apical membrane to the luminal surface of the ducts [33,34]. The non-dependence or non-responsiveness of tumors to androgens results in PSMA expression [35,36]. This cell-surface protein is overexpressed (approximately 1000 times greater than normal prostatic tissues) in most PCa (>90%) [37]. The level of PSMA expression is an important predictor for disease outcome [38,39]. PSMA is not specific only to the prostate gland, but it is expressed physiologically in normal cells in several organs, including salivary glands, lacrimal glands, proximal renal tubules, epididymis, the luminal side of the ileum-jejunum system, and central nervous system. It is also expressed by other cancers due to an overexpression of PSMA on cancer-related neovascular structures, such as the bladder, pancreas, lung, and renal cell cancers [28]. Although this antigen is not specific to PCa cells, it can be promoted as a target for imaging and treatment by reason of overexpression by tumors with potentially low impact on normal cells.

### 3.1. Anti-PSMA Antibodies

Indium-111 (^111^In) capromab-pendetide (ProstaScint, AYTU Bioscience, Englewood, NJ, USA) [40] was the first commercialized anti-PSMA antibody, which was approved by the US FDA in 1996. However, ProstaScint was only able to bind to the intracellular epitope of PSMA. It localizes to sites of cell membrane destruction (necrosis or apoptotic cell) that are nonviable cells. Image quality was affected by poor tumor penetration and high background activity resulting in suboptimal detection of pelvic nodal metastases during staging. Antibody-based approaches were, therefore, considered to have a limited diagnostic potential (delayed target recognition, low tumor-to-background ratios) [41].

### 3.2. PSMA Ligands for PET Imaging

The identification and characterization of a PSMA active substrate with its structural and functional homology to glutamate carboxypeptidase 2 (also known as N-acetylated-α-linked acidic dipeptidase I) contributed to the development of small molecules, including PSMA ligands and inhibitors. PSMA inhibitors are classified into three families: phosphorus-based (including phosphonate, phosphate, and phosphoramidate), thiol-based, and urea-based. PSMA PET radiotracer agent development emphasized small urea-based PSMA ligands which target PSMA’s extracellular active substrate recognition sites with high binding affinity to PCa cells, thus, showing rapid plasma clearance and high tumor-background ratios [42]. 

#### 3.2.1. Gallium-68 (^68^Ga)-Labeled PSMA Radiopharmaceuticals

The advance in development for the clinical application of PET imaging with PSMA ligands was reached with the formulation of ^68^Ga-PSMA-11 (also known as HBED-CC, HBED, PSMA-HBED, or Prostamedix), which exhibits useful characteristics. ^68^Ga-PSMA-11 or ^68^Ga-PSMA-HBED-CC was first reported in 2012 [43] and is now the most widely used radiopharmaceutical for PET imaging of the prostate. Due to the high level of binding and high accumulation intracellularly, ^68^Ga-PSMA-HBED-CC can potentially detect very small metastases. This agent is also rapidly cleared from non-target tissue. There is intense physiological uptake in the lacrimal and salivary glands. Moderate uptake in the liver, spleen, bowel, and sympathetic ganglia, such as cervical and coeliac ganglia, are seen and there is minimal uptake in normal prostate cells [44]. It is considered to be significantly useful in the diagnosis of recurrent PCa [45]. ^68^Ga-PSMA-11 has been proven to be highly sensitive in detecting disseminated PCa. In two studies of PCa patients with BCR, recurrent sites were detected in 90% of patients with elevated PSA [46,47]. ^68^GA-PSMA-11 PET/CT has a higher sensitivity compared to other PET agents, such as Choline [48,49,50]. This high sensitivity may have a major clinical impact to modify treatment plans and treatment modalities at different stages of PCa, from the primary diagnosis to during follow-up mCRPC after treatment. ^68^Ga-PSMA-11 PET/CT had a major impact on RT planning, leading to significant changes in treatment planning in more than 50% of patients. Furthermore, boost volumes of PET-positive LN were added in 80% of these cases [51]. ^68^Ga-PSMA-11 is mainly excretion via the kidneys with associated intense activity in the urinary tract. Focal activity in the ureters may be mistaken for pathological node uptake while bladder activity may interfere with prostatic bed assessment. Therefore, the focus shifted to the development of other radiotracers with little or slower urinary excretion. 

Simultaneous ^68^Ga-labeled PSMA PET/MRI improves the diagnostic accuracy for PCa localization compared with both mpMRI and with PET imaging alone [52]. The previous systematic reviews of Ling SW et al. revealed ^68^ that Ga-PSMA PET/MRI showed high diagnostic accuracy equivalent to that of ^68^Ga-PSMA PET/CT for the detection of extracapsular extension (ECE), seminal vesicle invasion (SVI), and LN metastases in primary staging of PCa [53]. There is an urgent need for research focused on the direct comparison of the two diagnostic tests. However, Hoffmann et al. suggested that ^68^Ga-PSMA PET/CT and PET/MRI are likely to become the standard imaging modalities in the staging of intermediate-to-high-risk primary PCa. Both methods have become the gold standard for restaging recurrent PCa in those countries and clinical centers where these imaging modalities are available. Their potential to facilitate more accurate prostatic biopsy, to guide therapy, and to monitor the therapeutic response to all treatment modalities is currently under intense investigation [54]. 

#### 3.2.2. Fluorine (^18^F)-Labeled PSMA Radiopharmaceuticals

Fluorine-18 has the advantage of a longer half-life. This enables centralized production and distribution over larger distances. The development of ^18^F-labeled PSMA compounds has led to a significant paradigm shift in the availability of PET imaging for primary and recurrent prostate cancers. This is primarily because there is a higher available amount of the radioisotope ^18^F, which is produced by a cyclotron, when compared with ^68^Ga, which is eluted from a generator. Excellent image quality is also the result of optimized tracer doses, better imaging statistics, and the decay properties of ^18^F itself. The first generation of ^18^F-labeled PSMA ligands was represented by ^18^F-DCFBC, described by Mease and colleagues [55]. The disadvantage of using this agent was the high background activity because slow blood clearance was found to interfere with the detection of LN metastases [56]. In 2011, the second generation of ^18^F-labeled PSMA ligands, ^18^F-DCFPyL, were introduced with satisfactory results due to its better image quality and small prostatic lesions visualized with excellent sensitivity [57]. ^18^F-DCFPyL is a promising alternative to ^68^Ga-PSMA-HBED-CC for PSMA-PET/CT imaging in recurrent PCa [58]. This ligand is characterized by rapid elimination via the urinary tract. However, neither ^18^F-DCFBC nor ^18^F-DCFPyL involve chelators capable of binding radionuclides for targeted therapy.

^18^F-PSMA-1007 performs at least comparably to ^68^Ga-PSMA-11, but its longer half-life with superior energy properties and non-renal excretion overcomes some limitations of ^68^Ga-labeled PSMA agents [59]. ^18^F-PSMA-1007 is a recently developed ^18^F-labeled PSMA PET radiopharmaceutical structurally related to PSMA-617 and includes a chelator capable of binding radionuclides for therapeutic purposes. The biodistribution is similar to that of other PSMA PET radiopharmaceuticals with the added advantage of minimal urine excretion due to predominant hepatobiliary excretion making this a very useful tool for more accurate pelvic nodal evaluation [59].

^68^Ga-PSMA-11 and ^18^F-DCFPyL are currently FDA-approved radioligands for PSMA-targeted PET imaging in PCa patients [60,61].

The molecular structure of PSMA and the effect of PSMA targeting radionuclide therapy to manage a PCa patient is shown in Figure 1.

## 4. Role of PSMA Imaging in PCa

### 4.1. PSMA Imaging for Initial Staging

Imaging has two major purposes at the initial diagnosis of PCa. Firstly, to assess the extent of disease in patients with biopsy-proven disease and high risk for metastases. Secondly, to localize primary tumors in patients with high suspicion and negative biopsies for PCa. Accurate staging has a considerable influence on guiding further loco-regional or systemic therapeutic options, such as radical prostatectomy, RT, or palliative treatment, and the extent of pelvic LN dissection during surgery or planning of the radiation field.

#### 4.1.1. T-Staging

MRI is currently the modality of choice for T-staging in patients with clinically suspected localized PCa with good diagnostic accuracy for cancer and ECE detection using the PIRADS [63]. Various protocols, for example, T2-weighted, dynamic contrast-enhanced (DCE) and diffusion-weighted sequences are performed to identify tumor extension, ECE, seminal vesicles and/or other organ invasion. Moreover, mpMRI can be combined with protocols to further distinguish between benign and malignant prostatic disease [64]. A major drawback of MRI is its variable diagnostic accuracy because of high interobserver variability for interpretation.

#### 4.1.2. N-Staging

Non-invasive imaging with CT and MRI may be used to detect LN metastases using LN diameter and morphology. LNs with a short axis of >8 mm in the pelvic region and >10 mm outside the pelvis are usually suspected of being malignant, however, the sensitivity of these techniques is only 36% while the specificity of these techniques is approximately 82% [8]. If there are pelvic node metastases in PCa patients, curative treatment by radical prostatectomy or radical RT is not the optimum treatment. Evaluation of LNs using CT or MRI is based on morphological information, and metastatic LNs are mainly detected based on increased size. However, nearly 80% of LN metastasis in PCa were smaller than the threshold size of 8 mm and were not usually detectable using morphological imaging [65].

From a meta-analysis by Perera M et al., the pooled sensitivity and specificity of ^68^Ga PSMA PET/CT for nodal staging are 75% and 99%, respectively [66]. Based on a systematic review, intermediate- or high-risk pre-treatment PCa, ^68^Ga-PSMA PET had a greater sensitivity and a slightly different specificity in detecting the LN metastases when compared to MRI. ^68^Ga-PSMA PET/CT was found to have a greater sensitivity of 65% compared with 41% of mpMRI and a comparable specificity of 94% compared with 92% for preoperative nodal staging in intermediate- and high-risk PCa [67].

The existing prospective single-center phase II study evaluating ^18^F-DCFPyL PET/CT in 25 patients found that the sensitivity and specificity for nodal metastasis were 71.4% and 88.9%, respectively, with 50% of involved nodes < 3 mm in size, and 3 patients (12%) had unsuspected distant metastasis [68]. A recent study of 20 PCa reported the sensitivity and specificity in detecting LN metastases as 39% and 100% with ^68^Ga-PSMA PET/CT, 8% and 100% with MRI/CT, and 36% and 83% with DW-MRI, respectively. True-positive nodal metastases on ^68^Ga-PSMA PET/CT typically sized between 9 and 11 mm in diameter while false-negative nodes had a median diameter of 4 mm [69]. The Pro-PSMA study in 2020 revealed that ^68^Ga PSMA PET/CT was 27% more accurate than conventional imaging (CI). They found a lower sensitivity (38% vs. 85%) and specificity (91% vs. 98%) for CI compared to PSMA PET/CT [70]. 

#### 4.1.3. M-Staging

Bone and visceral lesions of PCa that may be undetected using CI can be visualized by ^68^Ga-PSMA–PET [71,72]. BS is the most widely used method for assessing bone metastases from PCa, with combined sensitivity and specificity of 79% and 82%, respectively [73]. The previous study of Pyka et al. revealed that ^68^Ga-PSMA PET was better than planar BS for the detection of affected bone regions as well as determining overall bone metastasis in PCa. Sensitivity and specificity for the overall bone involvement were from 99–100% and from 88–100% for PET, and from 87–89% and from 61–96% (*p* < 0.001) for BS [74].

### 4.2. Evaluation of Biochemically Recurrent Disease (BCR)

Following radical treatment for PCa with either external beam RT (EBRT) or radical prostatectomy (RP), between 27 and 53% of patients experience BCR [75]. There are differences in BCR definitions between and within the main curative contexts. According to the recent EAU guidelines on PCa, the threshold for best predicting further metastases after RP is PSA > 0.4 ng/mL and higher [76,77,78]. Prospective studies have reported the advantages of PSMA-targeted imaging in BCR in acquiring useful clinical information that could eventually change therapeutic strategies [69,79,80]. Various international guidelines recommend PSMA-PET imaging to be considered to clarify equivocal findings, especially if the results will directly and immediately influence therapeutic decisions (Figure 2).

^68^Ga-PSMA PET/CT scans usually detect sites of previously unsuspected disease, and the impact was higher in patients with BCR. This highlights the key role of ^68^Ga-PSMA PET/CT in the treatment of PCa. In the previous systematic review of Perera M. et al., they revealed that the overall percentage of positive ^68^Ga-PSMA PET patients was 40% for primary staging and 76% for BCR. Positive ^68^Ga-PSMA PET scans for BCR patients increased with PSA before PET. The predicted positivity was 48% for PSA of 0.2 ng/mL, 56% for 0.5 ng/mL, and 70% for 1.0 ng/mL. Shorter PSA doubling time increased ^68^Ga-PSMA PET positivity [66]. Table 3 summarizes the role of PSMA PET/CT in PCa.

### 4.3. Clinical Interpretation and Common Pitfalls in PSMA-Targeted Imaging

PSMA PET imaging is a highly sensitive and specific method, but this is an evolving field and readers of scans need to be aware of a variety of physiological and other pathological reasons for the over-expression of PSMA to avoid interpretation errors. Multiple systems have been established for the interpretation of PSMA PET imaging. One system that has been considered is the PSMA reporting and data system (PSMA-RADS) version 1.0 [82]. PSMA-RADS is a system for approaching PSMA-targeted PET findings which are divided into 5 subcategories with higher numbers indicating a greater probability of PCa. PSMA-RADS-1 and PSMA-RADS-2 are either certainly or nearly certainly benign, respectively, while PSMA-RADS-4 indicates a high probability for PCa and PSMA-RADS-5 almost certainly represents PCa. PSMA RADS-3 indicates an indeterminate lesion and can include some abnormal features with or without radiotracer uptake that are unlikely to represent PCa. The most complex of the PSMA-RADS version 1.0 categories is PSMA-RADS-3, which is separated into 4 subcategories that consider either the uncertainty of the given lesion being compatible with PCa (PSMA-RADS-3A, PSMA-RADS-3B, and some PSMA-RADS-3D findings) or recommend the appearance of other cancers (PSMA-RADS-3C and some PSMA-RADS-3D). Of these subcategories, PSMA-RADS-3C (represents lesions with radiotracer uptake but the uptake is unlikely to be representative of PCa) and PSMA-RADS-3D (represents the suspected lesion for cancer but without uptake) are of significant importance. For PSMA-RADS-3C, lesions would be atypical for PCa but have high PSMA uptake and may represent a non-prostate cancer. With regards to PSMA-RADS-3D, these include lesions that are concerning for the presence of PCa or a non-prostate malignancy but lack radiotracer uptake. However, most cases with indeterminate lesions would be typical for PCa, such as LN (PSMA-RADS-3A) or bone lesions (PSMA-RADS-3B), but equivocal uptake and lack a correlative anatomical finding. As a highly sensitive imaging modality, one of the concerns is false-positive findings as this can influence subsequent patient management. 

#### 4.3.1. Bone Uptake

Bone is the most common site of distant metastasis in patients with advanced prostate cancer (up to 80%) [83]. False-positive PSMA uptake in bone may be associated with bone remodeling and increased vascularity. This can lead to misidentification of bone metastasis (Stage M1b) and can alter the subsequent treatment. Correlation with clinical history, the CT component of the PET/CT, or other imaging modalities often help to confidently differentiate bone metastases from benign bone disease. 


(a)Benign bone diseases


Paget’s disease, old fractures, and fibrous dysplasia have been occasionally found to have increased PSMA uptake [84,85,86]. A common cause of bone uptake is healing old fractures. Several cases have described increased ^68^Ga-PSMA-11 and ^18^F-DCFPyL uptake in healing fractures at multiple sites, such as vertebrae, sacrum, ribs, and distal radius [87,88] (Figure 3). The previous study of Mads Ryø Jochumsen [89] revealed increased ^68^Ga-PSMA uptake in rib fractures, characteristically found as “pearls on a string.” These are important pitfalls when reporting PSMA PET scans. Panagiotidis et al. also reported ^18^F-PSMA-1007 uptake in healing rib fractures with no other pathological findings. They concluded that readers of ^18^F-PSMA-1007 PET/CT scans should be aware of this potential pitfall, particularly those with a nontypical trauma pattern (such as solitary bone lesion) imitating bone metastases [90].

Paget’s disease has also been known to mimic bone metastases in PCa patients undergoing PSMA PET imaging. The disease is characterized by increased bone resorption followed by bone formation and increased bone vascularity [86]. Few reported cases revealed increased uptake at sacrum, ischial tuberosity, head of humerus, iliac, and pubic bone with CI or pathological findings consistent with Paget’s disease.


(b)Nonspecific bone uptake


Nonspecific bone uptake occurs in approximately 30% of patients who undergo ^18^F-PSMA-1007 PET/CT. Despite the abovementioned advantages of ^18^F-PSMA-1007 over ^68^Ga-PSMA-11, the previous studies found a high incidence of nonspecific bone uptake which presents clinical challenges in a large number of patients. The lower positron energy with higher spatial resolution, due to the longer half-life of ^18^F compared to ^68^Ga, were possible reasons for this higher incidence [91]. These are much more common with digital PET scanners rather than analog scanners. The most common site of nonspecific bone uptake is found in the ribs, followed by the pelvis and spine [92]. The previous study of Wang et al. reported bone metastatic distribution based on bone scans that found patients with fewer lesions tend to have metastases to the spine, followed by the pelvic bones [93]. They also reported that only 1% of patients had bone metastases outside the spine and pelvis without evidence of spine metastasis. This supports the theory that singular or multiple nonspecific bone uptake in the ribs without coexisting suspicious lesions in the spine or pelvis are most likely benign. Additionally, bone marrow islands, particularly in ribs, may be a cause of focal uptake (Figure 4). The previous cohort study of Chen et al. [94] revealed that solitary rib lesions were considered malignant only when their size was increased on follow-up imaging. A lesion was considered benign if the PSA persists <0.1 ng/mL after radical prostatectomy, <2 ng/mL above nadir after radiotherapy, histology was benign in rib biopsy or follow-up imaging showed no growth in rib lesion. The previous study of Arnfeld et al. suggested a SUVmax threshold 7.2 under which these lesions should be interpreted as likely benign [95]. The long-term impact of nonspecific bone uptake in ^18^F-PSMA-1007 PET has not yet been reported. 

#### 4.3.2. Uptake in Lymph Nodes

Lymph nodes are the second most common site of metastases in prostate cancer [83]^.^ The biodistribution of PSMA PET imaging shows physiological tracer uptake in the lacrimal and salivary glands, liver, spleen, kidneys, in some parts of the intestines, and in coeliac ganglia. In addition, we frequently observe radiotracer uptake in mediastinal LN. Although the uptake in such LNs is usually low, a clear differentiation between reactive nodes and metastatic nodes with low uptake can be challenging in some cases. 

PSMA-positive mediastinal benign LNs are seen in a significant proportion of patients. PSMA-positivity on the histopathological level was associated with the activation state of the LNs. Radiotracer may be accumulated within the mediastinal LNs via the lymphatic system of the lungs. LNs with follicular hyperplasia would more likely be visible on PSMA PET imaging, as they usually present with the strongest PSMA expression. This finding is not limited to mediastinal LNs. It can occur at all body locations, and activation of LNs could lead to PSMA-positive LNs. However, it is obvious that mediastinal LNs are frequently activated, as they are often exposed to inflammatory processes derived from the lungs. Usually, the PSMA-positive mediastinal LNs show a low uptake. Indeed, the quantitative analyses revealed that PSMA-positive mediastinal LNs show significantly lower tracer uptake compared to LN metastases. Previous studies have shown that in the patients who usually present with solitary PSMA-positive mediastinal LNs, none of the patients had more than 2 PSMA-positive mediastinal LNs (Figure 5). Contrarily, LN metastases, especially in more advanced disease, often present as a chain of clearly PSMA-positive lesions. This difference can also help to distinguish metastases from PSMA-positive LNs of benign origin. In cases of uncertain grading of PSMA-positive LN, it is recommended to conduct an additional late scan (for example, at 3 h post-radiotracer injection), and most LN metastases will be associated with increasing uptake over time, while most reactive mediastinal LNs demonstrate the opposite [96].

#### 4.3.3. Breast Uptake

Hormone-based therapies or androgen deprivation therapy (ADT) are being increasingly used in patients with PCa as less morbid alternatives to surgical castration. PCa patients who receive ADT appear to be at increased risk for gynecomastia with a prevalence as high as 75% [97]. Gynecomastia is characterized by abnormally enlarged male breasts by reason of hormonal imbalance (increase in estrogens at the expense of testosterone) [98]. It requires prompt recognition, evaluation, and management.

There are a few case reports about false-positive breast(s) uptake in PCa patients with gynecomastia. The case report of Sasikumar et al. [99] described false-positive uptake in bilateral gynecomastia on a ^68^Ga PSMA PET/CT scan in a PCa patient who received ADT. Ultrasound and mammography of this patient revealed no abnormal findings suggestive of malignancy in either breast. Similarly, the case report of Gozde Daglioz Gorur [100] showed that gynecomastia on ^68^Ga-PSMA PET/CT scans was correlated with moderate focal uptake in breast parenchyma. No pathological findings suggestive of metastasis or recurrence were observed. They concluded that gynecomastia may present with bilateral and symmetrical PSMA uptake, but unilateral or no PSMA uptake may also be seen on PET/CT scans in some atypical cases.

## 5. PSMA-Targeted Radionuclide Therapy

Androgen deprivation therapy (ADT) is usually a first-line option for men with advanced PCa, but most PCa patients undergo progression while receiving ADT, and this disease status is referred to as castration-resistant Pca (CRPC). The mechanisms driving progression from androgen-dependent (hormone-sensitive or castration-sensitive) Pca to CRPC are still largely unclear, although continued androgen receptor signaling is considered the underlying common factor. Men with advanced Pca who have evidence of progressive disease, such as an increase in serum PSA, new metastases, or progressive existing metastases during ADT, and who have castration levels of serum testosterone (<50 ng/dL), are considered to have CRPC. Nearly 90% of them present with bone metastases causing pain, fractures, and mortality. Multiple treatment modalities including ADT, chemotherapy, immunotherapy, and radionuclide therapy are promoted in the management of CRPC. Most of these men will be identified initially because of a rising serum PSA. Importantly, the presence of CRPC does not imply that the disease is totally independent of androgens and resistant to further therapies directed at blocking androgen stimulation.

### 5.1. PSMA-Targeted Radioligand Therapy

#### 5.1.1. ^177^Lu-PSMA Radioligand Therapy

Lutetium-177 (^177^Lu) is a beta-emitting radioisotope with a half-life of 6.7 days. It has an average energy of 133.6 keV with a maximum penetration depth of <2 mm [101]. Because of its good physical properties and the possibility of post-treatment imaging, ^177^Lu labeled PSMA has been extensively studied and it has emerged as a novel treatment for mCRPC. Until recently, the majority of PSMA-targeting tracers involve urea-based agents (small molecule inhibitors) including ^177^Lu-PSMA-I&T (imaging and therapy) and ^177^Lu-PSMA-617, with the latter being preferred due to lower renal uptake [102,103]. A previous study of Ruigrok et al. revealed that although ^177^Lu-PSMA-617 and ^177^Lu-PSMA-I&T show similar binding characteristics in prostate tumors, ^177^Lu PSMA-I&T has a lower tumor-to-kidney ratio than ^177^Lu PSMA-617 [104].

Protecting the kidneys and salivary and lacrimal glands is the major challenge for PSMA-targeting radiotracers due to high uptake in these organs. Salivary gland toxicity has been documented as dose-limiting. This significantly reduced the quality of life of treated patients. Recently, there have been more reports and publications focusing on the safety and efficacy of ^177^Lu-PSMA radioligand therapy. 

The systematic review and meta-analysis of Yadav et al. focused on 17 retrospective/prospective reports, totaling 744 patients treated with ^177^Lu PSMA-radioligand therapy and concluded that 75% of patients had a reduction of PSA levels, with 46% having a reduction of more than 50%. A radiographic partial remission was seen in 37%, median overall survival (OS) was 13.8 months, and median progression-free survival (PFS) was 11 months. The most common treatment-related side effects were myelosuppression, nephrotoxicity, and salivary gland toxicity (pain, swelling, and dry mouth) [21]. 


(a)^177^Lu-PSMA-617


The LuPSMA trial (prospective single-arm, single-center, phase II trial) was conducted between August 2015 and December 2016. Forty men with PSMA-avid metastatic CRPC were treated with ^177^Lu-PSMA-617 (7.5 GBq/cycle). Seventeen (57%) of thirty patients had a reduction in PSA of 50% or more. There were no treatment-related deaths. The most common adverse effect associated with ^177^Lu-PSMA-617 was grade 1 dry mouth (87%), and grade 3 or 4 thrombocytopenia that may have been caused by ^177^Lu-PSMA-617 occurred in 13%. Objective response in LN or visceral disease was found in 82% of patients. Median PSA progression was 7.6 months, and median OS was 13.5 months [105].

The TheraP study (randomized phase 2 clinical trial at 11 centers in Australia) was conducted between February 2018 and September 2019 with 200 mCRPC patients. Treatment in the study was given to 98 (99%) of 99 men randomly selected to ^177^Lu-PSMA-617 (6.0–8.5 GBq intravenously every 6 weeks for up to 6 cycles) compared to 85 (84%) of 101 randomly assigned to cabazitaxel (20 mg/m^2^). PSA responses were more common among men in the ^177^Lu-PSMA-617 group than in the cabazitaxel group. Grade 3–4 adverse events occurred in 32 of 98 men (33%) in the ^177^Lu-PSMA-617 group versus 45 of 85 men (53%) in the cabazitaxel group. No deaths were found in the ^177^Lu-PSMA-617 group [17].

The VISION study was a phase 3 randomized trial from June 2018 to mid-October 2019. The study enrolled 831 mCRPC patients with a median follow-up of 20.9 months. ^177^Lu-PSMA-617 (7.4 GBq every 6 weeks × 6 cycles) combined standard of care (SOC) compared to SOC alone and revealed significant improvement in OS by a median of 4.0 months and PFS based on imaging was significantly longer. The incidence of grade 3 or above adverse events was higher with ^177^Lu-PSMA-617 than the other arm but the quality of life was not affected. Due to the favorable treatment outcome with low incidence of adverse events in this study, the promotion of ^177^Lu-PSMA-617 as a standard protocol in advanced PSMA-positive mCRPC is suggested [13].

The LuTectomy trial (open label, phase 1/2, non-randomized clinical trial) evaluated the dosimetry, efficacy, and toxicity of ^177^Lu-PSMA in men with high PSMA-expressing high-risk localized or locoregional advanced PCa who underwent radical prostatectomy (RP) and pelvic lymph node dissection (PLND). This study started enrollment in August 2020 and is expected to be completed in June 2023 (NCT04430192) [106].

The UpFrontPSMA trial (open label, randomized, phase 2 trial) of sequential ^177^Lu-PSMA 617 and docetaxel Versus docetaxel alone in 140 newly diagnosed metastatic PCa. The objective of this study was to evaluate response to treatment by measuring PSA levels and radiological response and safety of ^177^Lu-PSMA 617. This study started enrollment in April 2020 and is expected to be completed in April 2024 (NCT04343885) [107].


(b)^177^Lu-PSMA-I&T


^177^Lu-labeled PSMA ligand (DOTAGA-(I-y) fk (Sub-KuE), also known as PSMA I&T, for “imaging and therapy”) is now considered essential for the treatment of advanced PCa [98]. PSMA-I&T was first developed in Germany in 2015. There are similar properties between ^177^Lu-PSMA-I&T and ^177^Lu-PSMA-617. In a study of 56 mCRPC patients who received an average dose of 5.76 GBq per cycle (total of 125 cycles), the PSA PFS was approximately 14 months with 59% of patients having >50% reduction in PSA levels [108]^.^

In a previous large cohort study in 2019 involving100 patients treated with a total 319 cycles of ^177^Lu-PSMA-I&T (median 2 cycles, range 1–6), 6–8 weekly with mean activity of 7.4 GBq, PSA decreased ≥50% within 12 weeks of treatment. Longer PFS and OS was observed in 38 patients, PFS was 4.1 mo and OS was 12.9 mo. Hematologic grade 3/4 toxicities were anemia (9%), thrombocytopenia (4%), and neutropenia (6%). Grade 3/4 non-hematologic toxicities were not found [109].

The SPLASH trial (a phase 3, open-label, randomized study) evaluated the efficacy of ^177^Lu PSMA-I&T (AKA. ^177^Lu-PNT2002) versus abiraterone or enzalutamide in slowing the progression of radiographic findings in mCRPC patients. The study will begin with a safety and dosimetry arm involving 25 patients (Part 1) and Part 2 involves a randomized therapeutic phase in 390 patients. All patients will have long-term follow-up for at least 5 years, death, or loss to follow-up (Part 3). This study began enrollment in March 2021 with approximately 415 patients (NCT04647526) [110].


(c)Combination of ^177^Lu-PSMA 617 with androgen receptor-axis-targeted therapies (ARAT)


ARAT was only approved for patients with metastatic castration. Systemic ARAT is now FDA-approved even in PCa without evidence of mCRPC. ARAT administration increases PSMA expression. In addition, ARAT may cause radiosensitization. It is hypothesized that combined ^177^Lu-PSMA radioligand treatment and ARAT might result in better tumor control in patients with CRPC [111].

The PSMAddition study is an international open-label, randomized, phase III study in 1126 metastatic hormone-sensitive PCa patients. The objective of this study was to evaluate the efficacy and safety of ^177^Lu-PSMA-617 combined with SOC, versus SOC alone. The SOC was determined as a combination of ARAT with ADT. Participants received approximately 7.4 GBq of ^177^Lu-PSMA-617, every 6 weeks for 6 planned cycles. The primary outcome was to assess radiographic PFS (rPFS) and estimated final OS analysis. This study started enrollment in June 2021 and is expected to be completed in December 2025 (NCT04720157) [112].

The ENZA-p trial (open label, randomized phase 2, multicenter) was to compare the efficacy and safety of 160 mg enzalutamide daily + 177Lu-PSMA-617 (up to 4 cycles of 7.5 GBq) versus enzalutamide alone in 160 mCRPC patients who were deemed at high-risk for early failure on enzalutamide monotherapy. This study started enrollment in August 2020 and is expected to be completed in June 2023 (NCT04419402) [113].

Even though there is a biological basis to promote combined ARAT and ^177^Lu-PSMA 617, key questions to answer will include the optimal dose, period of administration, and long-term safety of use of ^177^Lu-PSMA-617 in the early stage of subsequent lines of treatment. 


(d)Combination of ^177^Lu PSMA-617 with DNA damage repair inhibitor


Transcription active sites are often unstable and susceptible to breakage since the torsional stress and local depletion of nucleosomes allow DNA to be more accessible to damaging substances. A dedicated DNA damage response (DDR) is necessary to preserve genome integrity at the exposed sites. The DDR is a complex system of DNA damage sensor proteins, for example, the poly (ADP-ribose) polymerase 1 (PARP-1). It is important in repairing radiation-induced single-stranded DNA breaks, which permit cancer cells to become resistant to radiation [114]. Therefore, PARP-1 inhibition may cause radiosensitization when combined with radioligand treatment. PARP inhibitors, such as olaparib and rucaparib, have been described as effective against mCRPC [106,107]. 

LuPARP is an Australian phase 1, open-label, multicenter, dose-escalation and dose-expansion study to investigate the use of ^177^Lu-PSMA 617 with olaparib in 52 mCRPC patients who have progressed on novel AR targeted drugs and have PSMA-avid disease on imaging. Patients received fixed 7.4 GBq of ^177^Lu-PSMA-617 every 6 weeks plus olaparib. The primary endpoints were dose-limiting toxicity and maximum tolerated dose. Secondary endpoints were adverse events, rPFS, and OS. This study started enrollment in July 2019 and is expected to be completed in October 2022 (NCT03874884) [115].


(e)Combination of ^177^Lu PSMA-617 with checkpoint inhibitor immunotherapy


Immunotherapy has changed the therapeutic strategy of many hematological and solid cancers. However, several phase I and II trials evaluating programmed death receptor 1 (PD-1) and cytotoxic T-lymphocyte antigen-4 (CTLA-4) inhibitors have reported limited usage for mCRPC. Additionally, although sipuleucel-T represents the only FDA-approved cancer vaccine for mCRPC according to IMPACT trial results [116], its use is relatively limited in daily clinical practice. There are several ongoing clinical trials in various immunotherapy approaches either alone or combined with other therapies in mCRPC. Different mAb against PD-1, PD-L1, or CTLA-4 were tested for mCRP treatment with unsatisfactory results [117].

PRINCE is the phase Ib/II study of ^177^Lu-PSMA-617 combined with pembrolizumab for treatment in 37 mCRPC patients who have progressed on ARAT. Patients with PSMA-avid disease were enrolled to receive ^177^Lu-PSMA for up to 6 doses (starting at 8.5 GBq with a decrease of 0.5 GBq for each cycle) and pembrolizumab for up to 35 cycles (every 3 weeks). The primary objectives were PSA response rate and safety. Secondary objectives were rPFS, PSA-PFS, and OS. This study started recruitment in July 2019 and was completed in October 2021 (NCT03658447) [118].

#### 5.1.2. Alpha-Emitting PSMA-Targeted Radioligand Therapy

For mCRPC treatment, multiple studies have confirmed that ^177^Lu-PSMA-617 has a favorable dosimetry and good objective response, including improvement in PSA levels and radiological findings [119,120]. However, approximately 30% of patients did not respond to ^177^Lu-labeled PSMA ligands. Targeted α-radiation therapy may be more effective for mCRPC treatment and has been reported in a limited number of patients to be effective in patients resistant to ^177^Lu-PSMA-617 therapy. Despite having good tolerability, high radioactivity accumulations in bone metastases which lie near to or within the red marrow, indicate that the actual absorbed dose to some domains of active marrow may be rather higher than estimated in the previous study due to spillover resulting in associated developmental risk factor for hematologic toxicity. Recent studies have shown that targeted α-radiation therapy is highly beneficial for mCRPC patients in this setting [121]. 


(a)^225^Ac-PSMA-617


^225^Ac is an alpha emitter with a relatively long half-life (9.9 days) [122,123]. Actinium-225 (^225^Ac)-PSMA-617 has a significantly higher linear energy transfer (LET) compared to beta particles. With reference to a preliminary report from a single-center study, ^225^Ac-PSMA-617 (100 kBq/kg every two weeks) resulted in decreased PSA levels below measurable levels and showed complete response on imaging. No related hematologic toxicity was found. Xerostomia was the only clinical side effect mentioned [124]. In the previous study of Kratochwil et al. of fourteen PCa patients with metastasis, therapeutic activity of 100 kBq/kg of ^225^Ac PSMA-617 per cycle repeated every eight weeks demonstrated proper trade-off between toxicity and biochemical response. Low grade hematological toxicity was found in six patients and xerostomia was found in eight patients. Xerostomia was the dose-limiting factor with 100 kBq/kg considered the maximum tolerated dose [125]. A recent study of Sathekge M et al. assessed the therapeutic outcome of ^225^Ac-PSMA-617 in seventeen advanced PCa patients of which the results showed a good anti-tumoral effect assessed by serum PSA level and ^68^Ga-PSMA-PET/CT as seen in 94.1% of patients. A PSA decrease by ≥90% after treatment was found in 82.4% of patients. All patients experienced grade 1/2 xerostomia without severe symptoms [126]. Although targeted α-therapy with ^225^Ac-PSMA-617 is still considered experimental, it clearly has the potential to be of great benefit to advanced-stage PCa patients.


(b)^213^Bi-labeled PSMA-617


Bismuth-213 (^213^Bi) is a short half-life (45.6 min) mixed alpha and beta emitting agent [123,127]. Sathekge et al. reported a 1^st^ treatment case with ^213^Bi-PSMA-617 (two cycles with a cumulative activity of 592 MBq) in mCRPC patients who had progressed under conventional therapy gained PSMA-imaging response and biochemical response with a declined PSA level from 237 µg/L to 43 µg/L [128]. The previous study of Kratochwil et al. revealed that dosimetry of ^213^Bi-PSMA-617 is suitable for clinical application. However, when compared with ^225^Ac-PSMA-617, it suffers from higher perfusion-dependent off-target radiation, and a biological half-life of PSMA-617 in dose-limiting organs is longer than the physical half-life of ^213^Bi, making this agent a second-choice radiolabel for the targeted alpha therapy of PCa [129]. 

### 5.2. Anti-PSMA Radioimmunotherapy

J591 was the first monoclonal antibody (mAb) targeting the extracellular domain of PSMA that was derived from the original murine J591 (muJ591) by substituting epitopes of the B and T cell [130,131]. This led to differences in kinetics and biodistribution, the adverse events observed following two agents were often different. The prospective study of Tagawa concluded that the PSMA-targeted ^177^Lu antibody J591 was related with greater hematologic effects than with PSMA-617. However, ^177^Lu PSMA-617 is related with more non-hematological toxicity than ^177^Lu-J591. ^177^Lu-PSMA-617 was observed to be related with more PSA decline than ^177^Lu-J591, but no difference in OS was found on multivariable analysis [132]. The difference between mAb and Ligand are described in Table 4.

#### 5.2.1. ^177^Lu-J591 Antibody

J591 is typically labeled with lutetium-177 (^177^Lu-J591) to target PCa cells and deliver radioimmunotherapy. It has been studied extensively in phase I/II clinical trials. ^177^Lu-J591 was first studied in a phase I trial in 2005 in 35 mCRPC patients receiving up to three doses. Myelosuppression was a dose limited to 75 mCi/m^2^, and a dose level of 70-mCi/m^2^ was defined as the single-dose maximum tolerated dose. The biologic activity was observed in four patients with >50% reduction in PSA levels for between 3 and 8 months. No patient developed a human anti-J591 antibody response to deimmunized J591 [133]. In a phase 2 study of Tagawa et al., single doses of ^177^Lu-J591 were studied in forty-seven patients with progression after hormonal treatment. A total of 10.6% experienced ≥50% reduction in PSA, 36.2% experienced ≥30% decline, and 59.6% experienced any PSA decline after single treatment. Median OS was approximately 22 vs. 12 months in mCRPC treated with a single-dose of 70 mCi/m^2^ and 65 mCi/m^2^, respectively. However, a higher dose resulted in a higher grade of thrombocytopenia and neutropenia [134]. In a phase 1/2 study, forty-nine mCRPC patients received ^177^Lu-J591 doses ranging between 20 and 45 mCi/m^2^ × 2 two weeks apart. The recommended doses at phase 2 were 40 mCi/m^2^ and 45 mCi/m^2^ × 2. Median OS was 42.3 months vs. 19.6 months in patients receiving higher vs. lower doses. The higher grade of thrombocytopenia and neutropenia were found in higher doses than lower doses [135].

#### 5.2.2. ^225^Ac-J591 Antibody

The reason for higher hematological toxicity for ^177^Lu-J591 is possibly because mAb is a large molecule with slower circulating clearance compared to smaller molecules, for example, PSMA-617. However, compared to PSMA-617, J591 shows no uptake in the salivary gland or kidneys [136]. Because of this character, it has been hypothesized that ^225^Ac-J591 can reduce the incidence of salivary and renal toxicity found in ^225^Ac-PSMA-617. In the previous phase 1 trial of Tagawa, twenty-two mCRPC patients received seven dose levels of ^225^Ac-J591, and only one patient who received 80 KBq/kg had grade 4 thrombocytopenia and anemia. Xerostomia was seen only in patients receiving ^177^Lu-PSMA-617 treatment. PSA reduction was observed in 64% with 41% of patients having ≥50% PSA reduction (Clinical trial information: NCT03276572) [137].

#### 5.2.3. ^227^TH-PSMA-TTC Antibody

PSMA-Targeted Thorium-227 Conjugate (PSMA-TTC) is a targeted alpha therapy for PCa. Thorium-227 (^227^Th) is a long half-life (18.7 days) α-particle emitter which is the parent nuclide of radium-223. ^227^Th-PSMA-TTC is a novel human antibody attached to ^227Th^ and has shown very strong antitumor efficacy in animal models with PCa [138]. A phase I clinical trial of PSMA-TTC in mCRPC is currently ongoing based on promising preclinical data. It is planned with a recruitment of one hundred ninety-eight patients and is expected to be completed in September 2024 [139] (NCT03724747). Clinical trials of PSMA-targeted radioligand therapy is described in Table 5.

## 6. Economic Benefits and Cost-Effectiveness

There are several studies dealing with the potential cost-effectiveness of PSMA PET/CT [117,120]. The previous study of Scholte M et al. revealed that PSMA PET/CT and nano-MRI appear to be cost-effective compared to extended pelvic LN dissection (ePLND) because they save cost, but at the possible expense of a small quality-adjusted life years (QALY) loss [140].

The previous study of Gordon LG et al. [141] assessed the cost-effectiveness of ^68^Ga-PSMA PET/MRI for PCa with BCR. ^68^Ga-PSMA PET/MRI was compared with usual care in patients with suspected recurrent PCa. Thirty PCa with BCR from a study in Australia provided important model estimation. The primary outcomes were health system costs and life expectancy (survival) of more than 10 years. ^68^Ga-PSMA was estimated to cost AUD 56, 961 and produce 7.48 life years compared to AUD 64, 499 and 7.41 life years in usual care. Thereby, ^68^Ga-PSMA was potentially cost-saving (- AUD 7592) and slightly more effective 0.07 life years. In this exploratory economic assessment, the use of ^68^Ga-PSMA PET/MRI to detect recurrent PCa appears to be cost-effective compared to usual care.

The previous study of Cardet RF et al. [142] assessed the costs and diagnostic accuracy associated with using ^68^Ga-PSMA-11 PET/CT compared with CI in staging high-risk PCa use of information collected as part of the proPSMA study [70]. It revealed that the estimated cost per scan for ^68^Ga-PSMA-11 PET/CT was AUD 1203, which was less than the CI cost at AUD 1412. ^68^Ga-PSMA-11 PET/CT has better accuracy and lower cost. This resulted in a saving of AUD 959 per additional accurate nodal detection and saving of AUD 1412 for additional accurate distant metastases detection. A similar cohort-based analysis of Schwenck J et al. assessed consecutive ^68^Ga- PSMA PET/CT, ^11^C-choline PET/CT, and standard CT imaging in the same patient about TNM-staging and based on RT options. It also calculates the cost-efficacy of PET- versus CT-based treatment. They found concordant results in both PET studies (72% of patients), while the TNM staging concordant between ^68^Ga-PSMA PET and standard CT was only 36%. Incorrect staging results in “wrong” therapeutic decisions and, therefore, to additional costs. A ^68^Ga-PSMA PET study is cost-effective if additional costs do not exceed EUR 3844 (AUD 4312) compared to CT. They concluded that ^68^Ga-PSMA PET/CT is cost-effective in all patients considering the avoidance of incorrect treatment [143].

Although the published literature in the theranostics of PCa is substantial, there is limited information on comparative economic and true benefits of improved outcomes involving the whole pathway of improved diagnosis and targeted precision therapy. It is important that these improved diagnosis and treatment strategies should be incorporated into useful cost-effective tools to upgrade cancer management.

## 7. Conclusions

In recent years, PSMA has emerged as a promising target for imaging and therapy with radionuclides to support the principle of “theranostics” and promote the concept of Precision Medicine. PSMA-targeted imaging has been shown to be a very useful tool compared to cross-sectional imaging. While the detection of recurrent prostate cancer at even lower PSA levels will remain the main indication, several other potential clinical applications, including early disease detection leading to earlier treatment interventions and treatment approaches, need further attention. The role of PET-MRI in this setting is particularly interesting, and the idea of a one-stop test involving a PSMA PET ligand and an MRI scan is appealing. However, in spite of the widespread clinical use coupled with extensive research carried out so far, the overall impact on survival on the basis of PSMA-targeted imaging is still unclear. 

Additionally, PSMA targeted radionuclide therapy using beta/alpha emitters and associated benefits to progression free survival and overall survival presents a highly viable alternative method for treating patients with advanced prostate cancer. In the not too distant future, it is predicted and highly likely that PSMA based theranostics will become a standard of care in the management of patients with prostate cancer.

## Figures and Tables

**Figure 1 jcm-11-02738-f001:**
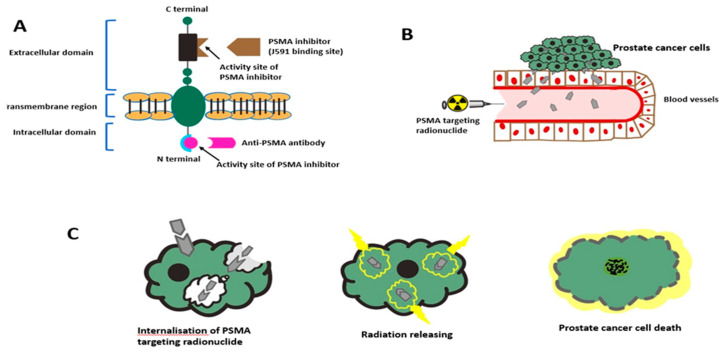
Schematic representation of PSMA molecular structure and effect of PSMA targeting radionuclide therapy to manage a PCa patient. (**A**) PSMA molecule has 3 domains; intracellular, transmembrane, and the large extracellular portions. (**B**) PSMA targeting radionuclide is injected into the bloodstream, the PSMA targeting radionuclide binds to the activity site on the prostate cancer cells. (**C**) After binding to the activity site on the cell membrane, the PSMA targeting radionuclide is internalized and releases radiation from within the cell. The end product is DNA damage with resultant tumor cell death. (Figure adapted from Mokoala K et al. in PSMA Theranostics: Science and Practice. Cancers 2021, 13, 3904 [62]).

**Figure 2 jcm-11-02738-f002:**
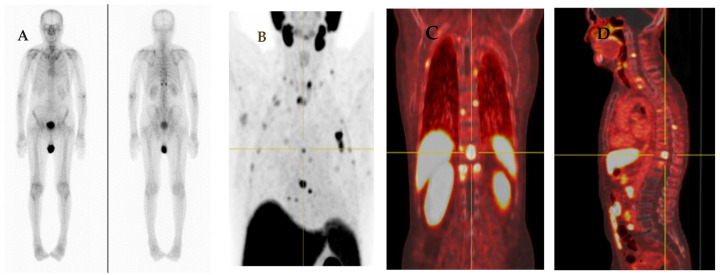
A 74-year-old man with adenocarcinoma of prostate gland (Gleason score 5 + 4 = 9) S/P radical prostatectomy with bilateral orchidectomy, developed rising serum PSA with level of 1.2 ng/mL which was suspected of BCR. His bone scan revealed equivocal lesions at left scapula, T8, and 12 vertebrae (**A**). He also performed ^18^F-PSMA PET/CT for evaluating BCR. There are multiple PSMA-avid lesions on MIP image (**B**) associated with multiple PSMA-avid mixed osteolytic and blastic metastases at multiple levels of vertebrae, both scapulae and multiple bilateral ribs as seen on coronal PET (**C**) and sagittal PET (**D**) images.

**Figure 3 jcm-11-02738-f003:**
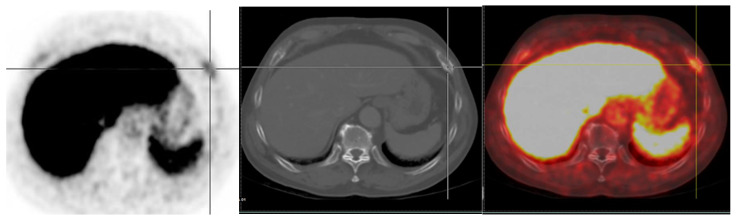
An example of old rib fracture in a PCa patient who underwent radical prostatectomy and RT, developed rising serum PSA with a level of 0.375 ng/mL (nadir 0.1 ng/mL) was sent to evaluate BCR. ^18^F PSMA PET scan revealed faint uptake at lateral aspect of left 7th rib associated with subtle sclerotic lesion (SUVmax of 3.14).

**Figure 4 jcm-11-02738-f004:**
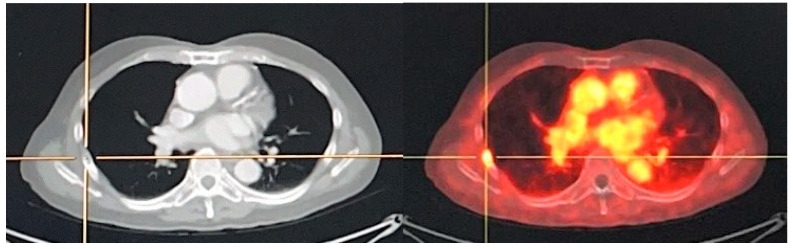
An example of nonspecific bone uptake. ^18^F-PSMA PET/CT scan performed on a 71-year-old male with a history of 3 + 3 Gleason score PCa treated with radical prostatectomy (initial PSA levels of 6.3 ng/mL) for initial staging. PET/CT scan revealed a focal PSMA uptake at lateral aspect of right 6th rib (SUVmax of 5.08) without associated osteolytic/blastic lesion. Pathological report from CT guided-biopsy at this lesion revealed benign bone lesion. Follow-up serum PSA in this patient was still low (nadir 0.006 ng/mL).

**Figure 5 jcm-11-02738-f005:**
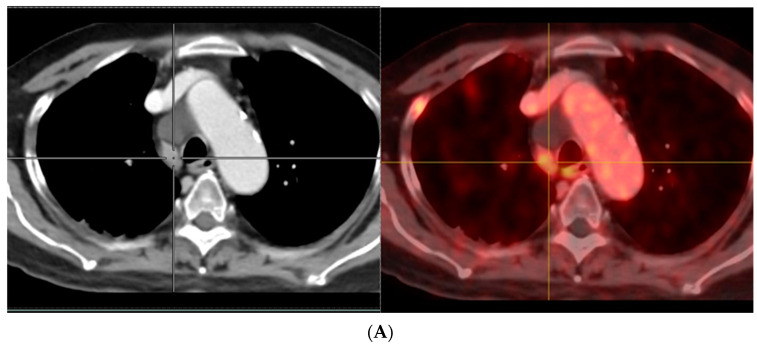
A 75-year-old man with diagnosed PCa (Gleason score of 4 + 4, initial PSA of 43.12 ng/mL) who underwent ^18^F-PSMA PET/CT for initial staging. PET/CT scan reveals PSMA uptake at prostatic bed with multiple osteoblastic metastases at vertebrae. There are few subcentimeter mild PSMA-avid nodes at right lower paratracheal regions (SUVmax of 2.72) which are suspected of reactive nodes (**A**). Transaxial PET/CT in lung window (**B**) revealed increased PSMA uptake associated with reticulonodular infiltration at RUL which was suspected of pulmonary infection. After follow-up imaging, pulmonary lesion was improved, and mediastinal nodes revealed no significant change in size but no longer observable PSMA uptake.

**Table 1 jcm-11-02738-t001:** EAU risk groups for biochemical recurrence of localized and locally advanced PCa.

Definition			
Low Risk	Intermediate Risk	High Risk	
PSA < 10 ng/mL GS < 7 or cT1-T2a	PSA 10–20 ng/mL GS = 7 or cT2b	PSA > 20 ng/mL GS > 7 or cT2c	Any PSA, any GS cT3-4, or cN+ Locally advanced

GS = Gleason score, PSA = prostate-specific antigen.

**Table 2 jcm-11-02738-t002:** Literature overview on PSMA-targeted radioligand therapy (RLT) in PCa.

Authors	Year	Type of Study	Objectives	Number of Studies and/or Patients	Results
Zhang et al. [12]	2021	Meta-analysis	To evaluate the clinical efficacy and safety of the ^177^Lu-PSMA-617 therapy in the treatment of metastatic castration-resistant prostate cancer (mCRPC).	12 studies, 508 patients	After the first cycle of treatment, the pooled rate of PSA decline was 69.30%, and that of >50% PSA decline was 35.90% without significant adverse events.
Sartor O, et al. [13]	2021	Prospective, open-label, randomized, international, phase 3 trial (VISION trial)	To compare efficacy of ^177^Lu-PSMA-617 (7.4 GBq every 6 weeks × 6 cycles) combined standard of care (SOC) compared to SOC alone	831 patients	Significant improvement in OS by median of 4.0 months and significantly longer PFS based on imaging.
Ballal et al. [14]	2021	Systematic Review	To evaluate the role of ^225^Ac-PSMA as a salvage treatment in mCRPC	3 studies, 141 patients	^225^Ac-PSMA-617 revealed biochemical response, improved survival, caused low treatment-related toxicity proving a promising salvage treatment option in mCRPC patients.
Sadaghiani MS et al. [15]	2021	Systematic Review	To evaluate the efficacy and toxicity of ^177^Lu-PSMA-targeted radioligand therapy (PRLT)	24 studies, 1192 patients	PRLT is associated with ≥50% reduction in PSA level in a large number of patients and a low rate of toxicity
Satapathy S et al. [16]	2021	Systematic Review	To evaluate the role of ^225^Ac-PSMA RLT in mCRPC.	10 studies, 256 patients	^225^Ac-PSMA RLT is an efficacious and safe treatment option for mCRPC.
Hofman MS et al. [17]	2021	Randomized, open-label, phase 2 trial (TheraP trial)	To compare ^177^Lu-PSMA-617 with cabazitaxel in patients with mCRPC.	291 patients	^177^Lu-PSMA-617 compared with cabazitaxel in Mcrpc led to a higher PSA response and fewer grade 3 or 4 adverse events.
von Eyben FE et al. [18]	2020	Systematic Review	To evaluate treatment outcome of ^177^Lu-PSMA RLT in mCRPC	36 studies, 2346 patients	Half of all patients obtained a PSA decline of ≥50% and lived longer than those with less PSA decline. 10% of developed hematologic toxicity (anemia grade 3)
Satapathy S et al. [19]	2020	Systematic review and meta-analysis	To evaluate the impact of visceral metastases on biochemical response and survival outcomes in mCRPC treated with ^177^Lu-PSMA RLT.	12 studies, 1504 patients	Presence of visceral metastases was associated with poor response and survival outcomes in patients of mCRPC treated with ^177^Lu-PSMA RLT
Kim YJ [20]	2020	Meta-analysis	To evaluate treatment responses after the 1st cycle of ^177^Lu-PSMA-617 RLT	10 studies, 455 patients	Two-thirds of any PSA decline and one-third of >50% PSA decline after the 1st cycle of ^177^Lu-PSMA-617 RLT in mCRPC. Any PSA decline showed survival prolongation after the 1st cycle of the ^177^Lu-PSMA-617.
Yadav MP et al. [21]	2019	Systematic Review and meta-analysis	To evaluate efficacy and safety data on ^177^Lu-PSMA RLT for mCRPC	17 studies, 744 patients	^177^Lu-PSMA RLT is an effective treatment of advanced-stage mCRPC refractory to SOC with low toxicity.
von Eyben FE et al^.^ [22]	2017	Systematic Review	To compare efficacy of ^177^Lu PSMA RLT and third-line treatment for mCRPC	12 studies, 669 patients	^177^Lu-PSMA-617 RTL and ^177^Lu-PSMA I&T gave better effects and caused fewer adverse effects than third-line treatment
Calopedos RJS et al. [23]	2017	Systematic review and meta-analysis	To assess treatment response of ^177^Lu-PSMA in mCRPC	10 studies, 369 patients	Two-thirds of patients had biochemical response (any PSA decline was 68%, >50% PSA decline was 37%)

**Table 3 jcm-11-02738-t003:** Summary role of PSMA PET/CT in PCa.

Role	Information
Diagnosis	No definite role of PSMA PET in diagnosis of PCa according to the European Association of Urology (EAU) guidelines; however, multiple studies have implied the potential role of PSMA PET/CT as a complementary modality with mpMRI in diagnosis of PCa. From the PRIMARY trial, patients could have avoided biopsy with positive mpMRI (PI-RADS ≥ 3) but negative PSMA PET/CT [81]. The validation study is expected to inform future clinical guidelines on the role of PSMA PET/CT in the diagnosis of PCa.
Primary staging	EAU recommended cross-sectional imaging of the abdomen including pelvis and bone scans for primary staging of intermediate-to-high risk PCa. o T-staging ▪ PSMA PET/CT is not currently recommended, mpMRI is the modality of choice in patients with clinically suspected localized PCa with good diagnostic accuracy for evaluation of tumor involvement and extraprostatic extension using the PIRADS system. o N-staging: ▪ PSMA PET/CT has a higher sensitivity and specificity than conventional imaging (CI), such as CT, because anatomical imaging relies primarily on size for detecting nodal metastases. ▪ A large proportion of nodal metastases in PCa (up to 80%) are smaller than 8 mm in size, deemed normal on CT. ▪ However, true-positive nodal metastases on PSMA PET/CT typically measured between 9 and 11 mm in diameter while false-negative nodes had a median diameter <4 mm. o M-staging: ▪ Bone and visceral lesions of PCa that are undetected using CI can be visualized by PSMA PET/CT. ▪ PSMA PET/CT has higher sensitivity and specificity than CI, such as bone scan. ▪ PSMA PET/CT has emerged as a powerful alternative to bone scan and CT in the staging of high-risk PCa with suspected metastases even at low PSA levels. The proPSMA trial demonstrated PSMA PET/CT had 27% greater accuracy than CI (92% vs. 65%) in identifying local and distant metastatic PCa with less radiation exposure. Useful in detection of primary tumor, nodal, bone, and visceral metastasis on a single image modality as “one-stop shop” with higher diagnostic accuracy than CI. Moreover, cost–effective analysis revealed that total cost per scan of PSMA PET/CT was cheaper than CI in complete staging for men with high-risk PCa [70].
Recurrent detection (re-staging)	Approximately between 28 and 53% of PCa patients experience biochemical recurrent (BCR), re-staging with imaging is recommended. The EAU guidelines have recently recommended performing PSMA PET/CT in BCR patients for appropriate active treatment. The detection rate of PSMA PET/CT in BCR significantly increased with higher serum PSA. The advantages of PSMA PET/CT in BCR in acquiring useful clinical information that could eventually change therapeutic strategies.
Selection for radionuclide therapy	To appropriately select patients for PSMA targeting radionuclide therapy (such as ^177^Lu-PSMA). Determine PSMA status, positive PSMA has shown potential in treatment of mCRPC patients and more likely to respond to PSMA targeting radionuclide therapy. Should be complementary with FDG PET/CT in advanced disease because PSMA expression may be lost in this group of patients. Useful for monitoring response to therapy (surgery, radiotherapy, chemotherapy, and radionuclide therapy)

**Table 4 jcm-11-02738-t004:** Differences between mAb and Ligands.

Ligand (PSMA 617)	mAb (J591)
Small (mw 1400)	Large (mw 150,000)
Short circulation time	Long circulation time (days)
optimal tumor imaging within hours	optimal tumor imaging at 3–8 days
Rapidly diffuse to all sites of expression	Mostly target via vasculature
Toxicities	Toxicities
Kidney Salivary glands Small intestine	Bone marrow Liver

**Table 5 jcm-11-02738-t005:** Clinical trials of PSMA-targeted radioligand therapy.

Clinical Trial	Status	Phase	Patients	Interventions
ACTRN12615000912583 (LuPSMA) [105]	completed	2	40	^177^Lu-PSMA-617 in progressive mCRPC
NCT03392428 (TheraP) [17]	active, not recruiting	2	200	^177^Lu-PSMA-617 vs. cabazitaxel in progressive mCRPC
NCT03511664(VISION) [13]	active, not recruiting	3	831	^177^Lu-PSMA-617 + SOC vs. SOC in progressive mCRPC
NCT04430192 (LuTectomy) [106]	recruiting	1/2	20	^177^Lu-PSMA-617 followed by prostatectomy
NCT04343885 (UpFrontPSMA) [107]	recruiting	2	140	Sequential ^177^Lu-PSMA-617 + docetaxel vs. docetaxel in metastatic hormone-naive PCa
NCT04419402 (ENZA-P) [113]	recruiting	2	160	Enzalutamide + ^177^Lu-PSMA-617 vs. Enzalutamide alone in mCRPC
NCT04647526 (SPLASH) [110]	recruiting	3	415	^177^Lu-PSMA-I&T vs. ARAT in progressive mCRPC
NCT04720157 (PSMAddition) [112]	recruiting	3	1126	^177^Lu-PSMA-617 + SOC vs. SOC alone in mHSPC
NCT03874884 (LuPARP) [115]	recruiting	1	52	^177^Lu-PSMA-617 + olaparib in progressive mCRPC
NCT03658447 (PRINCE) [118]	active, not recruiting	1/2	37	^177^Lutetium-PSMA-617 + pembrolizumab (mCRPC)
NCT03276572 [137]	active, not recruiting	1	31	^225^Ac-J591 in mCRPC
NCT03724747 [139]	recruiting	1	198	^227^Th-PSMA-TTC in progressive mCRPC

## Data Availability

Not applicable.

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
