# Peer review of "Update of PSMA Theranostics in Prostate Cancer: Current Applications and Future Trends"

_jcm, 2022, doi:10.3390/jcm11102738_

Round 1
Reviewer 1 Report
Review (Comments to the Author)
The review -Manuscript ID: jcm-1693339- entitled: "Update of PSMA theranostics in Prostate Cancer: Current applications and future trends" is valuable and well written.
However, there are major and minor aspects that should be considered.
Major changes
Line 32-34: The authors report that MRI and MRI-targeted biopsy are superior to standard biopsy. Studies should be added here in which the MRI-targeted biopsy is compared directly with the standard biopsy and the results are that standard biopsies should also continue to be carried out with the MRI-targeted biopsy, since the MRI-targeted biopsy alone often cannot find crucial foci. For this reason, the MRI-targeted biopsy has not yet completely replaced the standard biopsy, but there is usually a combination of both methods.
Line 53-54: The authors report that serum prostate-specific antigen (PSA) is
expressed by both normal and abnormal prostate cells. After that sentence they
report that is used as a biomarker for prostate cancer. The authors should explain in
which cases the expression of PSA is used as a biomarker and in which cases not
and place the differences for normal and abnormal prostate cells.
Line 60: The authors should use the actual reference for the EAU guidelines 2022:
"EAU Guidelines. Edn. presented at the EAU Annual Congress Amsterdam 2022. ISBN 978-94-92671-16-5.
EAU Guidelines Office, Arnhem, The Netherlands. http://uroweb.org/guidelines/compilations-of-all-guidelines/".
All recommendations of the EAU guidelines must also be adapted in the manuscript to the guidelines from 2022.
The authors should use the nomenclature in accordance with the International Consensus Radiochemistry Nomenclature Guidelines. After extensive discussion of
an international working group of experts in the field of radiochemistry and
radiopharmaceuticals, a consensus was reached and the recommendations for
nomenclature were published. The relevant literature is attached:
"Coenen HH, Gee AD, Adam M, Antoni G, Cutler CS, Fujibayashi Y, Jeong JM, Mach RH, Mindt TL, Pike VW, Windhorst AD. International Consensus Radiochemistry Nomenclature Guidelines. Nuklearmedizin. 2018 Feb;57(1):40-41. doi: 10.1055/s-0038-1636563. Epub 2018 Feb 21. PMID: 29536500".
The authors mention several times that accurate staging and restaging has influence on the guiding of therapy options (e.g. salvage radiotherapy in a biochemical setting). It would be recommended to add relevant references, such as "Hoffmann MA, Wieler HJ, Baues C, Kuntz NJ, Richardsen I, Schreckenberger M. The Impact of 68Ga-PSMA PET/CT and PET/MRI on the Management of Prostate Cancer. Urology. 2019 Aug;130:1-12. doi: 10.1016/j.urology.2019.04.004. Epub 2019 Apr 12. PMID: 30986486".
Minor changes
Line 31: Please correct "majol;r" in the introduction.
Line 35: In the introduction "PI-RADS V2" should be explained. This is discussed in
more detail on page 5 where it is shortened as "PIRADS". This is didactically
unfavorable. When it is mentioned for the first time, the terminology must already be
defined.
Line 80: The authors should correct "Eben F" to "von Eyben FE".
Line 161: Here is a text gap that should be corrected.
In line 232 the beginning of a sentence is missing!
Please correct the headline in line 270.
Line 274 explains the abbreviation EAU, which has already been mentioned on page
2.
In line 709 the authors should correct the wording!
Line 720: Here is a text gap that should be corrected.

Author Response
We thank you for your valuable suggestions. We’ve already revised according to your suggestions as described below
Major changes
1. Line 32-34: The authors report that MRI and MRI-targeted biopsy are superior to standard biopsy. Studies should be added here in which the MRI-targeted biopsy is compared directly with the standard biopsy and the results are that standard biopsies should also continue to be carried out with the MRI-targeted biopsy, since the MRI-targeted biopsy alone often cannot find crucial foci. For this reason, the MRI-targeted biopsy has not yet completely replaced the standard biopsy, but there is usually a combination of both methods.
Answer: Thank you for this suggestion. We agree with this suggestion and have added a sentence. Please see lines 36-41 (reference 5)
2. Line 53-54: The authors report that serum prostate-specific antigen (PSA) is expressed by both normal and abnormal prostate cells. After that sentence they report that is used as a biomarker for prostate cancer. The authors should explain in which cases the expression of PSA is used as a biomarker and in which cases not and place the differences for normal and abnormal prostate cells.
Answer: Thank you for this suggestion. We agree with this suggestion and have added a sentence. Please see lines 67-73.
3. Line 60: The authors should use the actual reference for the EAU guidelines 2022:
"EAU Guidelines. Edn. presented at the EAU Annual Congress Amsterdam 2022. ISBN 978-94-92671-16-5. EAU Guidelines Office, Arnhem, The Netherlands. http://uroweb.org/guidelines/compilations-of-all-guidelines/". All recommendations of the EAU guidelines must also be adapted in the manuscript to the guidelines from 2022.
Answer: Thank you for this suggestion. We agree with this suggestion and have added a sentence. Please see lines 805-808 (reference 11)
4. The authors should use the nomenclature in accordance with the International Consensus Radiochemistry Nomenclature Guidelines. After extensive discussion of an international working group of experts in the field of radiochemistry and radiopharmaceuticals, a consensus was reached and the recommendations for nomenclature were published. The relevant literature is attached: "Coenen HH, Gee AD, Adam M, Antoni G, Cutler CS, Fujibayashi Y, Jeong JM, Mach RH, Mindt TL, Pike VW, Windhorst AD. International Consensus Radiochemistry Nomenclature Guidelines. Nuklearmedizin. 2018 Feb;57(1):40-41. doi: 10.1055/s-0038-1636563. Epub 2018 Feb 21. PMID: 29536500".
Answer: Thank you for this suggestion. We agree with this suggestion and have revised the nomenclature appropriately.
5. The authors mention several times that accurate staging and restaging has influence on the guiding of therapy options (e.g. salvage radiotherapy in a biochemical setting). It would be recommended to add relevant references, such as "Hoffmann MA, Wieler HJ, Baues C, Kuntz NJ, Richardsen I, Schreckenberger M. The Impact of 68Ga-PSMA PET/CT and PET/MRI on the Management of Prostate Cancer. Urology. 2019 Aug;130:1-12. doi: 10.1016/j.urology.2019.04.004. Epub 2019 Apr 12. PMID: 30986486".
Answer: Thank you for this suggestion. We agree with this suggestion and have added a sentence. Please see lines 196-202 (reference 55)
For minor changes suggestions
1. Line 31: Please correct "majol;r" in the introduction.
Answer Please see line 33
2. Line 35: In the introduction "PI-RADS V2" should be explained. This is discussed in more detail on page 5 where it is shortened as "PIRADS". This is didactically unfavorable. When it is mentioned for the first time, the terminology must already be defined.
Answer Please see lines 42-43
3. Line 80: The authors should correct "Eben F" to "von Eyben FE".
Answer We’ve already removed it according to editor’s suggestion that suggested removing choline PET from the manuscript.
4. Line 161: Here is a text gap that should be corrected.
Answer We’ve already revised it.
5. In line 232 the beginning of a sentence is missing!
Answer We’ve already revised it.
6. Please correct the headline in line 270.
Answer We’ve already revised it.
7. Line 274 explains the abbreviation EAU, which has already been mentioned on page 2.
Answer We’ve already revised it.
8. In line 709 the authors should correct the wording!
Answer We’ve already revised it.
9. Line 720: Here is a text gap that should be corrected.
Answer We’ve already revised it.
Reviewer 2 Report
The present review entitled “Update of PSMA theranostics in Prostate Cancer: Current applications and future trends” aimed to outline the key concepts for Precision Medicine that contribute to improved prostate cancer diagnosis and focused therapy. The present review also intends to identify some essential insights and opportunities for additional research. Overall, the manuscript presents interesting data, however, some points need to be handled.
.
Key words (are missing)
This shall be stated after the abstract section.
INTRODUCTION
Line 21: Please consider “second” and “fifth” instead of “2nd” and “5th”.
Line 31: “majol;r".
Line 35-38: The information regarding the meta-analysis is not clearly presented. Please revise.
Line 42: The abbreviation “CT” is not previously written.
The introduction section needs to be revised in general; especially the final part, since the authors ended with the reference to table 1, which should be referred to in another section and not at the end of the introduction. As with other articles in JCM, the introduction should end with the main objective of this review. As example: Muselaers et al., 2022 (PSMA PET/CT in Renal Cell Carcinoma: An Overview of Current Literature).
Additionally, the reference 10 (Mottet et al., 2020) refers to the EAU risk group classification and not the European Society for Medical Oncology. According to Mottet et al., 2020 the guidelines on screening, diagnosis, and local treatment of clinically localised prostate cancer, refers to the European Association of Urology (EAU)-European Association of Nuclear Medicine (EANM)-European Society for Radiotherapy and Oncology (ESTRO)-European Society of Urogenital Radiology (ESUR)- International Society of Geriatric Oncology (SIOG) working group.
METHODS SECTION – Missing section
This is a crucial section that will improve the scientific value of the present review.
The author should add the Methods section after the Introduction. This is important to understand the search workflow and strategy of the present review.
Some point should be addressed, namely:
- Keywords
- Databases (for example: Pubmed and Web of Science, among others)
- Period of time
- Number of the studies
- Number of excluded studies
- Type of included studies (for example: larger cohort studies and case series)
- An overview of the included literature should be provided in a table, which could be a supplementary table.
GENERAL CONSIDERATIONS
- The authors should be encouraged to include a graphical abstract, represent the topic of the article in an attention-grabbing way.
- Acronyms/Abbreviations should be defined the first time they appear in each of three sections: the abstract; the main text; the first figure or table. When defined for the first time, the acronym/abbreviation/initialism should be added in parentheses after the written-out form.
- The authors cite four figures, however, and consider the article type, is not clearly understood the origin of this figures. Are they adapted from other articles? Are there a formal request for the proper use? This point should be carefully clarified.
- The presentation of titles and subtitles should be checked. The reader's attention can be lost throughout the text.
- Author Contributions and Conflicts of Interest are missing.
Overall, the article shall be reviewed and reworded in order to clarify the intended objective, although the complexity of the main issue.
Author Response
We thank you for your valuable suggestions. We’ve already revised according to your suggestions as described below
1. Key words (are missing) This shall be stated after the abstract section.
Answer We’ve already added them. Please see line 20
INTRODUCTION
1. Line 21: Please consider “second” and “fifth” instead of “2nd” and “5th”.
Answer Please see line 22
2. Line 31: “majol;r".
Answer Please see line 33
3. Line 35-38: The information regarding the meta-analysis is not clearly presented. Please revise.
Answer Please see lines 42-47 and lines 50-52
4. Line 42: The abbreviation “CT” is not previously written.
Answer Please see line 53
5. The introduction section needs to be revised in general; especially the final part, since the authors ended with the reference to table 1, which should be referred to in another section and not at the end of the introduction. As with other articles in JCM, the introduction should end with the main objective of this review. As example: Muselaers et al., 2022 (PSMA PET/CT in Renal Cell Carcinoma: An Overview of Current Literature).
Answer We’ve already revised it. Please see lines 81-91
6. Additionally, the reference 10 (Mottet et al., 2020) refers to the EAU risk group classification and not the European Society for Medical Oncology. According to Mottet et al., 2020 the guidelines on screening, diagnosis, and local treatment of clinically localised prostate cancer, refers to the European Association of Urology (EAU)-European Association of Nuclear Medicine (EANM)-European Society for Radiotherapy and Oncology (ESTRO)-European Society of Urogenital Radiology (ESUR)- International Society of Geriatric Oncology (SIOG) working group.
Answer We’ve already revised it. Please see line 77 and reference 11
- METHODS SECTION – Missing section
This is a crucial section that will improve the scientific value of the present review. The author should add the Methods section after the Introduction. This is important to understand the search workflow and strategy of the present review. Some point should be addressed, namely:
Keywords
Databases (for example: Pubmed and Web of Science, among others)
Period of time
Number of the studies
Number of excluded studies
Type of included studies (for example: larger cohort studies and case series)
An overview of the included literature should be provided in a table, which could be a supplementary table.
Answer We’ve already added it. Please see lines 93-105
GENERAL CONSIDERATIONS
1. The authors should be encouraged to include a graphical abstract, represent the topic of the article in an attention-grabbing way.
Answer We’ve already added it. Please see on Figure 1
2. Acronyms/Abbreviations should be defined the first time they appear in each of three sections: the abstract; the main text; the first figure or table. When defined for the first time, the acronym/abbreviation/initialism should be added in parentheses after the written-out form.
Answer We thank the reviewer for this comment and already revised it.
3. The authors cite four figures, however, and consider the article type, is not clearly understood the origin of this figures. Are they adapted from other articles? Are there a formal request for the proper use? This point should be carefully clarified.
Answer All of figures in this article are cases from our institution and I have removed the patient's personal data from showing in the images.
4. The presentation of titles and subtitles should be checked. The reader's attention can be lost throughout the text.
Answer We’ve already checked and revised them.
5. Author Contributions and Conflicts of Interest are missing.
Answer We’ve already added it. Please see lines 771-777
6. Overall, the article shall be reviewed and reworded in order to clarify the intended objective, although the complexity of the main issue.
Answer We’ve already reviewed and revised them.
Round 2
Reviewer 1 Report
no comments
Author Response
We thank the reviewer for your opinion.

Reviewer 2 Report
some points that should also be addressed: (1) Tables referring to the literature review should be presented as supplementary tables, given the extensive subject matter of the manuscript, it would be more appealing to the reader. (2) The references are not in line with the Instructions for authors, please check. (3) In Figure 2, 3, 4 and 5 should be presented the source of the material. The authors should identify the source of each figure. (4) The manuscript should be checked in order to correct some grammatical errors in the English.Author Response
We thank the reviewer for all of your comments and already revised it as described below
Some points that should also be addressed:
1. Tables referring to the literature review should be presented as supplementary tables, given the extensive subject matter of the manuscript, it would be more appealing to the reader.
Answer. Thank you for this suggestion. This is agreed and actioned.
2. The references are not in line with the Instructions for authors, please check.
Answer Thank you for this suggestion. This is agreed and actioned.
3. In Figure 2, 3, 4 and 5 should be presented the source of the material. The authors should identify the source of each figure.
Answer Thank you for this suggestion. This is agreed and actioned.
4. The manuscript should be checked in order to correct some grammatical errors in the English.
Answer Thank you for this suggestion. This is agreed and actioned. Our senior co-author “Professor Sobhan Vinjamuri” Head of nuclear medicine department, The Royal Liverpool University Hospital, who is a native-speaker and specialist in the article content, have proof read this manuscript before re-submitting.
